# Multi-dataset Training of Transformers for Robust Action Recognition

**Junwei Liang**[1]*, **Enwei Zhang**[2], **Jun Zhang**[2], **Chunhua Shen**[3]
[1]AI Thrust, Hong Kong University of Science and Technology (Guangzhou)
[2]Tencent Youtu Lab
[3]Zhejiang University
Email: junweiliang@hkust-gz.edu.cn, miyozhang@tencent.com,
bobbyjzhang@tencent.com, chunhua@me.com

## Abstract

We study the task of robust feature representations, aiming to generalize well on multiple datasets for action recognition. We build our method on Transformers for its efficacy. Although we have witnessed great progress for video action recognition in the past decade, it remains challenging yet valuable how to train a single model that can perform well across multiple datasets. Here, we propose a novel multi-dataset training paradigm, `MultiTrain`, with the design of two new loss terms, namely informative loss and projection loss, aiming to learn robust representations for action recognition. In particular, the informative loss maximizes the expressiveness of the feature embedding while the projection loss for each dataset mines the intrinsic relations between classes across datasets. We verify the effectiveness of our method on five challenging datasets, Kinetics-400, Kinetics-700, Moments-in-Time, Activitynet and Something-something-v2 datasets. Extensive experimental results show that our method can consistently improve state-of-the-art performance. Code and models are available at `https://github.com/JunweiLiang/MultiTrain`

## 1 Introduction

Human vision can recognize video actions efficiently despite the variations of scenes and domains. Convolutional neural networks (CNNs) [48, 49, 6, 44, 19, 36] effectively exploit the power of modern computational devices and employ spatial-temporal filters to recognize actions, which considerably outperform traditional models such as oriented filtering in space-time (HOG3D) [30]. However, due to the high variations in space-time, the state-of-the-art accuracy of action recognition is still far from being satisfactory, compared with the success of 2D CNNs in image recognition [24].

Recently, vision transformers such as ViT [15], and MViT [17] that are based on the self-attention [52] mechanism were proposed to tackle the problems of image and video recognition, and achieved impressive performance. Instead of modeling pixels as CNNs, transformers apply attentions on top of visual tokens. The inductive bias of translation invariance in CNNs makes it require less training data than attention-based transformers in general. In contrast, transformer has the advantage that it can better leverage 'big data', leading to improved accuracy than CNNs. We have witnessed a rapid growth in video datasets [28] in recent years, which would make up for the shortcomings of data-hungry transformers. The video data has not only grown in quantity from hundreds to millions of videos [42], but also evolved from simple actions such as handshaking to complicated daily activities from the Kinetics-700 dataset [7]. Meanwhile, transformers combined with low-level convolutional operations have been proposed [17] to further improve the efficiency and accuracy.

---

*Corresponding author. This work was partially done when JL was with Tencent.

36th Conference on Neural Information Processing Systems (NeurIPS 2022).

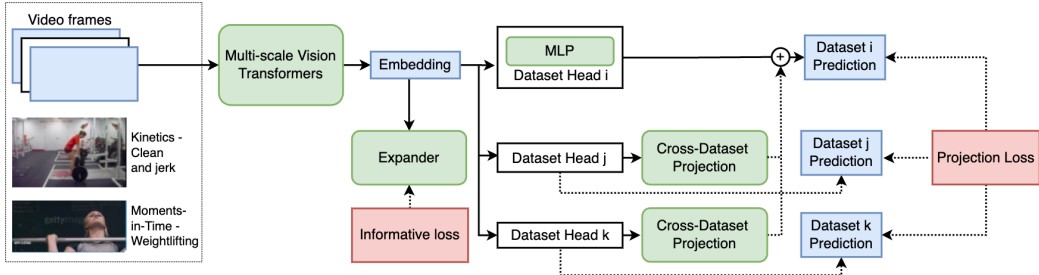

**Figure 1:** Overview of our multi-dataset training framework. We propose to utilize the intrinsic relations between classes across different action datasets. As we can see, the two video examples from Kinetics and Moments-in-Time datasets, respectively, show that samples from these two classes can be used to train both classification heads. The videos from multiple action datasets are input to the MViTv2 (Section 3.1) backbone and the model is trained jointly. The informative loss is applied to maximize the information content of the embedding from the backbone and the projection loss is applied to learn the intrinsic relations (Section 3.2).

Due to the data-hungry nature of transformers, most transformer-based models for action recognition requires large-scale pre-training with image datasets such as ImageNet-21K [14] and JFT-3B [58] to achieve good performance. This pre-training and fine-tuning training paradigm is time-consuming and it is not parameter-efficient, meaning that for each action dataset, a new model need to be trained end-to-end. Different from large image datasets such as ImageNet-21K that covers a wide range of object classes, currently the most diverse action dataset, Kinetics-700, only contains 700 classes. Each action dataset may be also limited to a certain topic or camera views. For example, Moments-in-Time [42] only contains short actions that happen in three seconds and Something-Something-v2 [23] focuses on close-up camera view of person-object interactions. These dataset biases might hinder models trained on a single dataset to generalize and be used in practical applications. These challenges in action datasets make learning a general-purpose action model very difficult. An ideal model should be able to cover a wide range of action classes meanwhile keeping the computation cost low. However, *simply combining all these datasets to train a joint model does not lead to good performance* [38]. In previous work [59], the authors have shown the benefit of training a joint model using multiple action datasets but their method requires large-scale image datasets such as ImageNet-21K [14] and JFT-3B [58], which is not available to the research community.

In this paper, we propose a general training paradigm for **Multi**-dataset **Train**ing of robust action recognition models, `MultiTrain`. Our method is designed to learn robust and informative feature representations in a principled approach, using the informative loss for regularization. We do not assume the availability of large-scale image datasets pre-training (although one can certainly take advantage of that). Since there are intrinsic relations between different classes across different action datasets (See Fig. 1 for examples of similar classes from two datasets), we propose a projection loss to mine such relations such that the whole network is trained to avoid over-fitting to certain dataset biases. Finally, all proposed loss terms are weighted using learned parameters. Thus, no hyper-parameter tuning is needed. Our empirical findings as shown in Table 1 indicate that our robust training method can consistently improve model backbone performance across multiple datasets. We show that our model can achieve competitive results compared to state-of-the-art methods, even without large-scale image dataset pre-training, and with a lower computational cost.

The main contributions of this paper are thus three-fold:

- To our knowledge, this is the first work to introduce informative representation regularization into multi-dataset training for improving action recognition.

- We propose an effective approach to mine intrinsic class relations in multi-dataset training by introducing the projection loss.

- Our method requires negligible computation overhead during training and no additional computation during inference to the backbone network. Extensive experiments on various datasets suggest our method can consistently improve performance.

## 2 Related Work

We review some work that is closest to ours.

**CNNs and Vision Transformers.** CNNs work as the standard backbones throughout computer vision tasks for image and video. Various effective convolutional neural architectures have been raised to improve the precision and efficiency (*e.g.*, VGG [45], ResNet [24] and DenseNet [26]). Although CNNs are still the primary models for computer vision, the Vision Transformers have already shown their enormous potential. Vision Transformer (ViT [15]) directly applies the architecture of Transformer on image classification and gets encouraging performance. ViT and its variants [2, 37, 4, 17, 40, 54] achieve outstanding results in both image and video processing in recent years.

**Action Recognition/Classification.** The research of action recognition has advanced with both new datasets and new models. One of the largest modern benchmarks for action recognition is the Kinetics dataset [28]. The Kinetics dataset proposes a large benchmark with more categories and more videos (*e.g.*, 400 categories 160,000 clips in [28] and 700 categories in [7]) as more challenging benchmarks compared to previous datasets like UCF-101 [47]. The Moments-in-Time [42] (MiT) dataset provides a million short video clips that cover 305 action categories. Note that it is infeasible for Kinetics and MiT datasets to cover all the possible actions in all possible scales. For example, surveillance actions [10, 8] are missing in the two datasets. Many new approaches [50, 60, 39, 20, 55, 36, 10, 8, 27, 9, 35] have been carried out on these datasets, of which the SlowFast network [20] and MViT [17] obtain promising performance. We can see that the trend of action recognition in the last two decades is to collect larger datasets (*e.g.*, Kinetics) and build models with a larger capacity.

**Multi-dataset Co-Training.** Previously, multi-dataset co-training has been explored in the image domain such as detection [62, 53] and segmentation [32]. Several works [43, 11, 46, 25] were proposed to combine multiple video datasets for training. Larger datasets often deliver better results. Combining multiple datasets to boost data size, and improve the final performance [22], and the simultaneous use of multiple datasets is also likely to alleviate the damaging impact of dataset bias. OmniSource [16] utilizes web images as part of the training dataset to expand the diversity of the training data to reduce dataset bias. VATT [1] uses additional multi-modal data for self-supversied pretraining and finetunes on downstream datasets. CoVeR [59] combines image and video training even during the finetuning stage and reports significant performance boost compared to single-dataset training. PolyViT [38] further extends to training with image, video and audio datasets using different sampling procedures. In this paper, we propose a simple yet effective way (no multi-stage training, no complex dataset-wise sampling and hyper-parameter tuning) for multi-action-dataset training, without the use of any image or additional data from other modality.

**Video Domain Generalization (VDG).** Our work is also related but different from video domain generalization [61]. The key distinction is that our goal is to train a single model on multiple related tasks (multiple action datasets) such that the model performs well on the same set of tasks, whereas VDG aims to generalize a model to unseen out-of-distribution target domain [56, 11–13, 51]. These models still suffer from problem of parameter-inefficiency, meaning that separate models are needed for different target datasets.

## 3 Method

Our method is built upon the backbone of the improved Multi-scale Vision Transformers (MViTv2) [34, 17]. Note that our approach works with any action recognition backbones. Given videos from multiple datasets during training, the model backbone takes the video frames and produces feature embeddings for each video. The same number of Multi-layer Perceptron (MLP) as the datasets are constructed as model heads to predict action classes for each dataset. To facilitate robust cross-dataset training, we propose two loss terms, namely, *the informative loss and projection loss*. The informative loss aims to maximize the embeddings' representative power. The projection loss, with the help of multiple cross-dataset projection layers, guides the model to learn intrinsic relations between classes of different dataset, hence the model heads can be trained jointly. See Fig. 1 for an overview of our framework. In this section, we first briefly describe the MViTv2 backbone design, and then present our proposed robust cross-dataset training paradigm.

### 3.1 The MViTv2 Backbone

Our model is based on the improved multi-scale vision transformers (MViTv2) [17, 34], which learns a hierarchy from dense (in space) and simple (in channels) to coarse and complex features. The series of work of vision transformers [15] (ViTs) follows the basic self-attention architecture [52] originally proposed for machine translation. The key component of the MViTv1 model [17] is the Multi Head Pooling Attention (MHPA), which pools the sequence of latent tensors to reduce the spatial or temporal dimension of the feature representations. In MViTv2 [34], a residual connection in MHPA for the pooled query tensor and a decomposed relative position embedding [2] are added. In this paper, we use 3D convolution as the pooling operation. Please refer to supplemental material for a visualization of the MViTv2 block. Each MViTv2 block consists of a multi-head pooling attention layer (MHPA) and a multi-layer perceptron (MLP), and the residual connections are built in each layer. The feature of each MViTv2 block is computed by:

$$
\begin{aligned}
X_1 &= \mathrm{MHPA}(LN(X)) + Pool(X) \\
Block(X) &= \mathrm{MLP}(LN(X_1)) + X_1,
\end{aligned}
\tag{1}
$$

where $X$ is the input tensor to each block. Multiple MViTv2 blocks are grouped into stages to reduce the spatial dimension while increase the channel dimension. The full backbone architecture is listed in supplementary material.

**Classification head.** For the action recognition problem, the model produces C-class classification logits by first averaging the feature tensor from the last stage along the spatial-temporal dimensions (we do not use the [CLASS] token in our transformer implementation), denoted as $\mathbf{z} \in \mathbb{R}^d$. A linear classification layer is then applied on the averaged feature tensor to produce the final output, $\mathbf{y} = \mathbf{W}_{\mathrm{out}}\mathbf{z} \in \mathbb{R}^C$.

**Multi-dataset training paradigm.** In general, to facilitate multi-dataset training of $K$ datasets, the same number of classification heads are appended to the feature embeddings. The $k$-th dataset classification output is defined as $\mathbf{Y}_k = h_k(\mathbf{Z}; \mathbf{W}_k) \in \mathbb{R}^{B \times C}$, where $h_k$ can be a linear layer or a MLP and $\mathbf{W}_k$ is the layer parameter.

### 3.2 MultiTrain: Robust Multi-dataset Training

Our training process fully leverages different action recognition datasets by enforcing an **informative loss** to maximize the expressiveness of the feature embedding and a **projection loss** for each dataset that mines the intrinsic relations between classes across other datasets. We then use uncertainty to weight different loss terms without the need for any hyper-parameters.

**Informative loss.** Inspired by the recently proposed VICReg [3] and Barlow Twins [57] methods for self-supervised learning in image recognition, we propose to utilize an informative loss function with two terms, **variance and covariance**, to maximize the expressiveness of each variable of the embedding. This loss is applied to each mini-batch, without the need for batch-wise nor feature-wise normalization. Given the feature embeddings of the mini-batch, $\mathbf{Z} \in \mathbb{R}^{B \times d}$, an expander (implemented as a two-layer MLP) maps the representations into an embedding space for the informative loss to be computed, denoted as $\mathbf{Z}' \in \mathbb{R}^{B \times d}$. The **variance loss** is computed using a hinge function and the standard deviation of each dimension of the embeddings by:

$$
\mathcal{L}^v = \frac{1}{d} \sum_{j=1}^{d} \max\left(0, 1 - \sqrt{\frac{\sum(\mathbf{Z}'_{ij} - \bar{\mathbf{Z}}'_{:j})}{d-1} + \epsilon}\right),
\tag{2}
$$

where : is a tensor slicing operation that extracts all elements from a dimension, and $\bar{\mathbf{Z}}'_{:j}$ is the mean over the mini-batch for $j$-th dimension. $\epsilon$ is a small scalar preventing numerical instabilities. With random sampling videos across multiple datasets for each batch, this criterion encourages the variance of each dimension in the embedding to be close to 1, preventing embedding collapse [57].

---

[2]We did not implement this part as the code were not available at the time of writing (March 2022).

The **covariance loss** $c(\mathbf{Z}')$ is defined as:

$$C(\mathbf{Z}') = \frac{1}{n-1}\sum_{i=1}^{n}(\mathbf{Z}'_i - \bar{\mathbf{Z}}')(\mathbf{Z}'_i - \bar{\mathbf{Z}}')^T \ , \ \text{where } \bar{\mathbf{Z}}' = \frac{1}{n}\sum_{i=1}^{n}\bar{\mathbf{Z}}'_i$$

$$\mathcal{L}^c = \frac{1}{d}\sum_{i\neq j}[C(\mathbf{Z}')]^2_{i,j} \tag{3}$$

Inspired by VICReg [3] and Barlow Twins [57], we first compute the covariance matrix of the feature embeddings in the batch, $C(\mathbf{Z}')$, and then define the covariance term $\mathcal{L}^c$ as the sum of the squared off-diagonal coefficients of $C(\mathbf{Z}')$, scaled by a factor of $1/d$.

**Projection Loss.** In previous works [59, 38], the intrinsic relations between classes from across different datasets have been mostly ignored during training. We believe that samples in one dataset can be utilized to train the classification head of other datasets. As shown in Fig. 1, the "Clean and jerk" video sample from Kinetics can be considered as a positive sample for "Weightlifting" in Moments-in-Time as well (but not vice versa). Based on this intuition, we propose to add a directed projection layer for each pair of datasets for the model to learn such intrinsic relations. One can also initialize the projection using prior knowledge but it is out-of-scope for this paper. Given the output from the $k$-th dataset classification output, the projected classification output is defined as:

$$\mathbf{Y}'_k = \mathbf{Y}_k + \sum_{i\neq k}^{K-1}\mathbf{W}^{proj}_{ik}\mathbf{Y}_i \in \mathbb{R}^{C_k}, \tag{4}$$

where $C_k$ is the number of classes for the $k$-th dataset and $\mathbf{W}^{proj}_{ik}$ is the learned directed class projection weights from $i$-th to $k$-th dataset. In this paper we only consider a linear projection function. We then use the ground truth labels of the $k$-th dataset to compute standarad cross-entropy loss:

$$\mathcal{L}_k = -\sum_{c=1}^{C_k}\hat{\mathbf{Y}}_{k,c}\log(\mathbf{Y}'_{k,c}), \tag{5}$$

where $\hat{\mathbf{Y}}_{k,c}$ is the ground truth label for the $c$-th class from the $k$-th dataset.

**Training.** We jointly optimize the informative loss and the projection loss during multi-dataset training. To avoid tuning loss weights of different terms, we borrow the weighting scheme from multi-task learning [29] and define the overall objective function as:

$$\mathcal{L}(\sigma) = \mathcal{L}^v + \mathcal{L}^c + \sum_{k=1}^{K}\frac{1}{2\sigma_k^2}\mathcal{L}_k + \log\sigma_k, \tag{6}$$

where $\sigma$ is a vector of learnable parameters of size $K$ (the number of datasets) for each projection loss term. This avoids the need to manually tune loss weights for different datasets.

## 4 Experiments

In this section, to demonstrate the efficacy of our training framework, we carry out experiments on five action recognition datasets, including Kinetics-400 [28], Something-Something-v2 [23], Moments-in-Time [42], Activitynet [5] and Kinetics-700 [7]. The action recognition task is defined to be a classification task given a trimmed video clip. Unlike previous works [38, 59], we do not initialize our model using ImageNet [14] since it consumes more computation. Please refer to the supplementary material for detailed comparison between train from scratch recipe and from ImageNet. In the experiments, we aim to showcase that our method can achieve significant performance improvement with minimal computation overhead compared to baselines.

### 4.1 Experimental Setup

**Datasets.** We evaluate our method on five datasets. Kinetics-400 [28] (K400) consists of about 240K training videos and 20K validation videos in 400 human action classes. The videos are about 10 seconds long. Kinetics-700 [7] (K700) extends the action classes to 700 with 545K training and

**Table 1:** Comparison with state-of-the-art on Kinetics-400, Moments-in-Time, Something-something-v2 and ActivityNet. We divide the baselines into two groups based on whether they are parameter-efficient. We report top-1/top-5 accuracy for each dataset. The **bold** numbers and underlined are ranked first and second, respectively. The FLOPs computation is for a single video clip input. Training data: (a) ImageNet-21K; (b) ImageNet-1K; (c) AudioSet [21]; (d) HowTo100M [41]; (e) Kinetics-400; (f) SSv2; (g) MiT; (h) ActivityNet.

| Method | Pre-/Training Data | gFLOPs | K400 | MiT | SSv2 | ActNet |
|---|---|---|---|---|---|---|
| ViViT [2] | +(a) | 3992 | 81.3 / 94.7 | 38.5 / 64.1 | 65.9 / 89.9 | - |
| VidTr [33] | +(a) | 392 | 80.5 / 94.6 | - | 63.0 / - | - |
| TimeSFormer [4] | +(a) | 2380 | 80.7 / 94.7 | - | 62.4 / - | - |
| X3D-XXL [18] | +(a) | 194 | 80.4 / 94.6 | - | - | - |
| MoViNet [31] | Scratch | 386 | 81.5 / 95.3 | 40.2 / - | 64.1 / 88.8 | |
| MViT-B [17] | Scratch | 455 | 81.2 / 95.1 | - | 67.7 / 90.9 | - |
| MTV-B (320p) [54] | +(a) | 1116 | 82.4 / 95.2 | 41.7 / 69.7 | 68.5 / 90.4 | - |
| Video Swin [40] | +(a) | 2107 | 84.9 / 96.7 | - | 69.6 / - | - |
| MViTv2-L [34] | +(a) | 2828 | **86.1 / 97.0** | - | **73.3 / 94.1** | - |
| MViTv2 w/ abs. pos. | Scratch | 225 | 80.4 / - | - | - | - |
| Ours-baseline | Scratch | 224 | 79.8 / 93.9 | 38.6 / 67.5 | 67.0 / 90.7 | 81.5 / 95.1 |
| VATT [1]* | (c),(d), Downstream | 2483 | 82.1 / 95.5 | 41.1 / 67.7 | - | - |
| CoVER [59] | (a),(e),(f),(g) | 2380 | 83.1 / - | 41.3 / - | 64.2 / - | - |
| PolyViT [38] | (b),(e),(f) + [Audio] + [Image] | 3992 | 82.4 / 95.0 | 38.6 / 65.5 | - | - |
| MultiTrain | (e),(f),(g),(h) | 224 | 81.9 / 95.2 | 41.7 / 71.0 | 68.9 / 91.6 | 87.4 / 97.3 |
| MultiTrain (312p) | (e),(f),(g),(h) | 614 | 83.2 / 96.4 | **43.1** / 71.9 | 69.3 / 92.1 | 88.2 / 97.6 |

35K validation videos. The Something-Something-v2 (SSv2) [23] dataset contains person-object interactions, which emphasizes temporal modeling. SSv2 includes 168K videos for training and 24K videos for evaluation on 174 action classes. The Moments-in-Time (MiT) dataset is one of the largest action dataset with 727K training and 30k validation videos. MiT videos are mostly short 3-second clips. The ActivityNet dataset [5] (ActNet) originally contains untrimmed videos with temporal annotations of 200 action classes. We cut the annotated segments of the videos into 10-second long clips and split the dataset into 107K training and 16K testing. Following previous works [20, 59], we follow the standard dataset split and report top-1/top-5 classification accuracy on the test split for all datasets. We conduct two sets of experiments, namely, "K400, MiT, SSv2, ActNet", and "K700, MiT, SSv2, ActNet".

**Implementation.** Our backbone model utilizes MViTv2 as described in Section 3.1. Our models are trained from scratch with random initialization, without using any pre-training (same as in [20] and different from previous works [59, 38] that require large-scale image dataset pre-training like ImageNet-21K [14] or JFT-3B [58]). We follow standard dataset splits as previous works [34, 20, 54]. See more details in the supplementary material.

**Baselines.** PolyViT [38] utilizes multi-task learning on image, video and audio datasets to improve vision transformer performance. The backbone they used are based on ViT-ViViT [2]. Similarly, VATT [1] utilizes additional multi-modal data for self-supversied pretraining and finetunes on downstream datasets. The backbone network is based on ViT [15]. CoVER [59] is a recently proposed co-training method that includes training with images and videos simultaneously. Their model backbone is based on TimeSFormer [4]. We also compare our method with other recent models trained using large-scale image datasets. See Table 1 and Table 2 for the full list.

## 4.2 Main Results

We summarize our method's performance in Table 1 and Table 2. We train our model jointly on MiT, SSv2, ActNet and two versions of the Kinetics datasets.

**Table 2:** Comparison with state-of-the-art on Kinetics-700, Moments-in-Time, Something-something-v2 and ActivityNet. The **bold** numbers and underlined are ranked first and second, respectively. See text and caption in Table 1 for details. Training data: (a) ImageNet-21K; (c) AudioSet [21]; (d) HowTo100M [41]; (e) Kinetics-700; (f) SSv2; (g) MiT; (h) ActivityNet.

| Method | Training Data | gFLOPs | K700 | MiT | SSv2 | ActNet |
|---|---|---|---|---|---|---|
| VidTr [33] | +(a) | 392 | 70.8 / - | - | 63.0 / - | - |
| MoViNet [31] | Scratch | 386 | 72.3 / - | - | - | - |
| MTV-B (320p) [54] | +(a) | 1116 | 75.2 / 91.7 | 41.7 / 69.7 | 68.5 / 90.4 | - |
| MViT-v2 [34] | +(a) | 2828 | **79.4 / 94.9** | - | **73.3 / 94.1** | - |
| Ours-baseline | Scratch | 224 | 74.1 / 91.9 | 38.6 / 67.5 | 67.0 / 90.7 | 81.5 / 95.1 |
| VATT [1]* | (c),(d), Downstream | 2483 | 72.7 / - | 41.1 / 67.7 | - | - |
| CoVER [59] | (a),(e),(f),(g) | 2380 | 74.9 / - | 41.5 / - | 64.7 / - | - |
| MultiTrain | (e),(f),(g),(h) | 224 | 75.8 / 93.2 | 42.2 / 72.3 | 69.1 / 92.2 | 88.1 / 97.2 |
| MultiTrain (312p) | (e),(f),(g),(h) | 614 | 76.3 / 93.5 | 43.5 / 73.0 | 70.4 / 93.1 | 89.1 / 98.1 |

We first compare our method with the original MViTv2 backbone in Table 1. "MViTv2 w/ abs. pos." means MViTv2 model with absolute positional embedding, which is taken from Table A.6 of MViTv2 paper [34] and it is (almost) the same as our model implementation. We can not achieve the same accuracy with the same recipe as MViTv2, which may be due to differences in the Kinetics dataset (missing some videos, etc. See supplementary material for full dataset statistics). PolyViT [38] is trained jointly with multiple image, audio and video datasets. We list the larger ones. We train our baseline model on the training set of each dataset to investigate the baseline performance. As we see, after adding robust joint training proposed in this paper, performance on each dataset has increased by 2.1%, 3.1%, 1.9% and 5.9% on K400, MiT, SSv2, ActivityNet, respectively in terms of top-1 accuracy. Note that our method achieves such improvement withtout large-scale image pre-training and additional inference computational cost.

We then compare our method with state-of-the-art on these datasets. We train a higher resolution model with larger spatial inputs (312p) and achieves better performance compared to recent multi-dataset training methods, CoVER [59] and PolyVit [38], on Kinetics-400, and significantly better on MiT and SSv2, as shown in Table 1. Note that our model does not use any image training datasets, and our model computation cost is only a fraction of the baselines. We also show that our performance boost does not come from the additional training dataset of ActivityNet in Table 3.

Our method also achieves competitive results compared to state-of-the-art models trained with large-scale image dataset (ImageNet-21K [14]). Compared to a recent method, MTV-B [54], our method is able to achieve significantly better top-1 accuracy across Kinetics-400, MiT, SSv2 by 0.8%, 1.4%, 0.8%, respectively, at half of the computation cost and without large-scale pre-training. Note that our model is parameter-efficient, while multiple MTV-B models need to be trained and tested on these datasets separately. Our method can achieve better performance with a deeper base backbone or larger resolution inputs but we have not tested due to limitation of computation resources.

We then compare our method on the Kinetics-700, MiT, SSv2 and ActivityNet training with baselines. Our parameter-efficient model can achieve better performance than MTV-B [54] at one-fifth of the computation cost. With a larger resolution model at 312p, we achieves significantly better performance than the baseline across Kinetics-400, MiT, SSv2 by 2.2%, 4.9%, 3.4%, respectively.

### 4.3 Ablation Experiments

In this section, we perform ablation studies on the K400 set. To understand how action models can benefit from our training method, we explore the following questions (results are shown in Table 3):

**Does our proposed robust loss help?** We compare our model training with vanilla multi-dataset training, where multiple classification heads are attached to the same backbone and the model is trained simply with cross-entropy loss. The vanilla model is trained from a K400 checkpoint as ours. As shown in Table 3, we try training the vanilla model with both the same training schedule as ours and a 4x longer schedule. As we see, there is a significant gap between the overall performance of the

**Table 3:** Ablation experiments. We investigate the effectiveness of each component of our method as well as compare to vanilla multi-dataset training method. "Vanilla" means using cross entropy (CE) loss in training. "w/o informative los" means using CE and projection loss. The numbers are top-1/top-5 accuracy, respectively. Training data: (e) Kinetics-400; (f) SSv2; (g) MiT; (h) ActivityNet.

| Method | Training Data | K400 | MiT | SSv2 | ActNet |
|---|---|---|---|---|---|
| MultiTrain | (e),(f),(g),(h) | 81.9 / 95.2 | 41.7 / 71.0 | 68.9 / 91.6 | 87.4 / 97.3 |
| Vanilla (50 ep) | (e),(f), (g),(h) | 80.1/ 94.0 | 33.4 / 60.1 | 60.8 / 89.0 | 86.5 / 97.1 |
| Vanilla (200 ep) | (e),(f), (g),(h) | 80.6 / 94.7 | 35.1 / 63.9 | 56.8 / 85.3 | 86.3 / 97.2 |
| w/o Informative Loss | (e),(f),(g),(h) | 13.5 / 33.4 | 7.3 / 19.9 | 9.7 / 28.5 | 24.8 / 54.3 |
| w/o Informative Loss & w/o Projection Add | (e),(f),(g),(h) | 80.4 / 94.5 | 38.7 / 68.9 | 62.6 / 89.8 | 86.5 / 97.4 |
| w/o Projection Loss | (e),(f),(g),(h) | 80.6 / 94.8 | 39.9 / 69.2 | 61.5 / 88.0 | 86.9 / 97.5 |
| w/o ActNet | (e),(f),(g) | 81.4 / 95.0 | 41.3 / 70.5 | 68.7 / 91.3 | - |

**Table 4:** Visualization of projection weights between different action datasets. We output the top-5 final cross-dataset projection layer weights of the K400-312p model. For all cross-dataset pairs, please refer to supplementary materials.

| Cross-dataset Pair | Top-5 Action Pairs |
|---|---|
| K400 to MiT | [('bending metal', 'bending'), ('riding elephant', 'skating'), ('pushing wheelchair', 'swinging'), ('tossing coin', 'tattooing'), ('cleaning toilet', 'plunging')] |
| K400 to ActNet | [('parkour', 'Capoeira'), ('running on treadmill', 'Walking the dog'), ('whistling', 'Snowboarding'), ('water sliding', 'Kayaking'), ('riding mule', 'Canoeing')] |
| SSv2 to K400 | [('Wiping something off of something', 'cleaning windows'), ('Throwing something in the air and catching it', 'dining'), ('Pretending or trying and failing to twist something', 'playing poker'), ('Turning the camera upwards while filming something', 'playing poker'), ('Pulling two ends of something so that it separates into two pieces', 'dining')] |
| MiT to K400 | [('gambling', 'bookbinding'), ('autographing', 'eating ice cream'), ('tearing', 'ripping paper'), ('hitchhiking', 'throwing ball'), ('sneezing', 'sneezing')] |

vanilla model and ours, validating the efficacy of our proposed method. Also, longer training schedule does not lead to better performance on some datasets, including SSv2, suggesting vanilla multi-dataset training is unstable. In terms of performance on ActivityNet, we observe that both training methods achieve good results, which might be because ActivityNet classes are highly overlapped with Kinetics-400 (65 out of 200).

**How important is the informative loss?** We then experiment with removing the informative loss (Section 3.2) during multi-dataset training. It seems that the feature embedding of the model collapse and the model is not trained at all. We further investigate why "w/o informative Loss" completely fails but "Vanilla" seems to work by running an experiment of "w/o informative Loss & w/o projection add", which means we remove the projected logits addition in Eq. 5 and directly compute classification loss on the projected logits. Therefore we can consider this run as adding additional projection branches to the vanilla architecture. The results are slightly better than "Vanilla" on K400 and much better on MiT/SSv2. It indicates that adding projected logits to the original branch without informative loss would prevent the model from converging (the total loss does not go down).

**How important is the projection loss?** We then experiment with removing the projection heads (Section 3.2) during multi-dataset training. The model is trained with the original cross-entropy loss and the informative loss. As shown in Table 3, the performance on MiT and SSv2 suffers by a large margin, indicating that the projection design helps boost training by better utilizing multi-dataset information.

**What does the cross-dataset projection layer learn?** We analyze the cross-dataset projection weights of the K400/312p model and list top 5 concepts for each pair of datasets in Table 4. We make

two observations. First, the top projections are visually similar actions, which confirms our intuition that there are intrinsic relations in the datasets that the model can mine to improve performance. For example, "bending metal" in K400 and "bending" in MIT, "parkour" in K400 and "Capoeira" in Activitynet. Interestingly, "Wiping something off of something" in SSv2 and "cleaning windows" in K400. Second, the action with the same name may not have the highest weights. In "mit to kinetics", the "sneezing" action ranks 5th in the projection weights, suggesting that there might be discrepancies of the same concept in different datasets. These observations are interesting and one may compare the learned weights with textual semantic relations (like those in ConceptNet). We leave this to future work.

**Does the additional ActivityNet data help?** In previous methods like CoVER and PolyViT, the ActivityNet dataset has not been used. In this experiment, we investigate the important of the ActivityNet dataset by removing it from the training set. From Table 3, we can see that the performance across all datasets drop by a small margin, indicating our superior results compared to CoVER (see Table 1 and Table 2) come from the proposed robust training paradigm rather than the additional data.

### 4.4 Discussion

By multi-dataset training transformers on various datasets, we obtain competitive results on multiple action datasets, without large-scale image datasets pre-training. Our method, `MultiTrain`, is parameter-efficient and does not require hyper-parameter tuning. Current limitations of our experiments are that we have not tried co-training with image datasets such as ImageNet-21K [14]. Hence we do not know how much performance gain that would entail. We plan to explore this in future work. In addition, we have not tried training larger model with FLOPs on par with state-of-the-art or other backbone architectures (*e.g.*, CNNs) due to limitation of our computational resources. Hence we are not sure how our algorithm would behave with these models. We have not explicitly explored how temporal modeling could benefit from multi-action-dataset training, which we leave for future work. Although our model is trained on multiple datasets, potential dataset biases can still cause negative societal impact in real-world deployment, as the datasets we have do not fully represent all aspects of human actions.

## 5 Conclusion

In this paper, we present `MultiTrain`, a robust multi-dataset training approach that maximizes information content of representation and learns intrinsic relations between individual datasets. Our method can train parameter-efficient models that perform well across multiple datasets.

## 6 Acknowledgement

This work was in part supported by Foshan HKUST Projects (FSUST21-FYTRI01A, FSUST21-FYTRI02A). C. Shen's participation was in part supported by a major grant from Zhejiang Provincial Government.

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
