# Supplementary Material: Multi-dataset Training of Transformers for Robust Action Recognition

**Junwei Liang**[1][*]**, Enwei Zhang**[2]**, Jun Zhang**[2]**, Chunhua Shen**[3]
[1]AI Thrust, Hong Kong University of Science and Technology (Guangzhou)
[2]Tencent Youtu Lab
[3]Zhejiang University
Email: junweiliang@hkust-gz.edu.cn, miyozhang@tencent.com,
bobbyjzhang@tencent.com, chunhua@me.com

36th Conference on Neural Information Processing Systems (NeurIPS 2022).

In this paper, we present ***MultiTrain***, a robust multi-dataset training approach that maximizes information content of representation and learns intrinsic relations between individual datasets. In this supplementary material, we present more details about the backbone we used and the experiments.

## 1 The MViTv2 Backbone

The full MViTv2 architecture is listed in Table 2. In contrast to natural language which can be directly tokenized into words, given the input video $\mathbf{V} \in \mathbb{R}^{T \times H \times W \times 3}$, ViTs extract tokens by splitting the video into $N = \lfloor T/t \rfloor \times \lfloor H/h \rfloor \times \lfloor W/w \rfloor$ non-overlapping patches, $\{\mathbf{v_1}, \cdots, \mathbf{v_N} \in \mathbb{R}^{t \times h \times w}\}$. Each patch is then projected into a patch embedding by a 3D convolution operator $E$. All patch embeddings are then concatenated into a sequence, and separate learnable spatial-temporal positional embeddings $\mathbf{p_s}, \mathbf{p_t}$ are also added to this sequence. The patch embedding process is denoted by:

$$\mathbf{X_0} = [\mathbf{Ev_1} \cdots \mathbf{Ev_N}] + P(\mathbf{p_s}, \mathbf{p_t}) \in \mathbb{R}^{N \times d_p} \tag{1}$$

The $P$ function extends the separate position embedding into the length of the sequence by repeating at the same spatial or temporal location. $d_p$ is the dimension of the patch embedding.

See the main text for MViTv2 block design. A visualization of the block design is shown in Fig. 1

---

[*]Corresponding author. This work was partially done when JL was with Tencent.

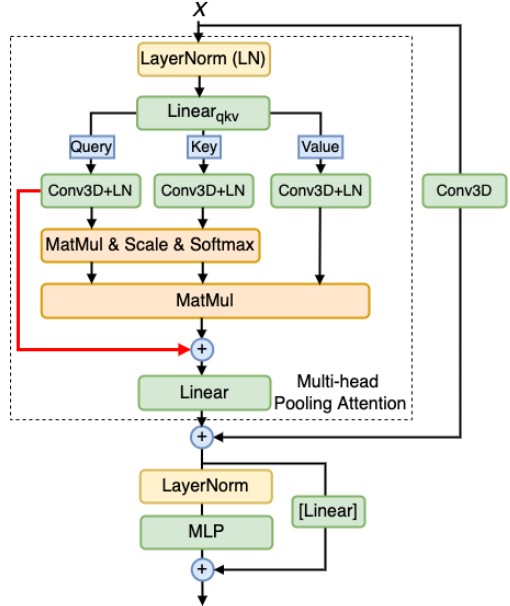

Figure 1: The MViTv2 Block. The residual connection for pooled query tensor (red arrow) and the residual 3D convolution operation outside the Multi-head Pooling Attention block are additions to the MViTv1 [3] design. The linear layer in the residual connection of the MLP block is only needed when the output embedding dimension is different. Compared to the MViTv2 paper [5], we do not use the decomposed relative embedding.

## 2 Experiments

### 2.1 Implementation Details

For both sets of experiments, we start our training from the corresponding Kinetics-trained model checkpoints to reduce training time. The Kinetics pre-training follows "train from scratch" protocol of MViTs [3, 5]. We use a learning rate of 2e-4 and cosine learning rate decay. We use synchronized SGD with 128 V100/A100 GPUs to train for 50 epochs with a batch size of 512, with a linear warm-up from 2e-6 for the first 10 epochs. Each experiment takes 2-4 days to finish. We use a weight decay of 0.05 and the adamw [6] optimizer is used. We use 32x3 video frames (number of frames times sampling rate) as inputs except for SSv2. For SSv2, following the MViT paper [3], we uniformly sample frames from the whole videos instead of getting fix-length clips. For all three datasets, we use random $224 \times 224$ crops unless mentioned otherwise and horizontal flipping from a video clip (except for videos from SSv2), which is randomly sampled from the full-length video and resized to a shorter edge side of randomly sampled in [256, 320] pixels. We utilize Mixup [9] and Rand Augment [1] for data augmentation during training. Following previous works [4, 3], we sample $3 \times 3$ (unless mentioned otherwise) clips for each video during testing: we uniformly sample 3 clips for the temporal domain and 3 spatial crops of size $224 \times 224$ after the shorter edge size are resized to 256 pixels. We average the softmax scores across all clips for final prediction. We utilize 2-layer MLPs with 2048 hidden size and a GELU activation for the expander and the classification heads. A single-layer linear projection head is added for each pair of datasets during training, and they are discarded during inference (the original classification head outputs are used as the final prediction.)

### 2.2 Dataset Statistics

For the Kinetics-400 and Kinetics-700 datasets, we have to download the original videos from Youtube. The difference in performance compared to original paper's implementation of MViTv2 could be due to the small difference in the datasets (Kinetics videos are taking down from Youtube from the time of release. See Table 1 for full statistics.

Table 1: The number of available videos from Youtube for each split at the time of our experiments compared to the original dataset on release.

| Dataset | Train | Validation | Released |
|---|---|---|---|
| Kinetics 400 | 234583 (95.3%) | 19760 (98.8%) | May 2017 |
| Kinetics 700 | 525460 (96.4%) | 33370 (95.3%) | July 2019 |

Table 2: The MViTv2 structure. See the main text for the MViTv2 Block design.

| Stage | Operators | Output Size |
|---|---|---|
| Data | stride $3 \times 1 \times 1$ | $3 \times 32 \times 224 \times 224$ |
| Patch | Conv3D $3 \times 7 \times 7$, 96, stride $2 \times 4 \times 4$ | $96 \times 16 \times 56 \times 56$ |
| $Stage_1$ | MViTv2 Block $\times$ 2 | $96 \times 16 \times 56 \times 56$ |
| $Stage_2$ | MViTv2 Block $\times$ 3 | $192 \times 16 \times 28 \times 28$ |
| $Stage_3$ | MViTv2 Block $\times$ 16 | $384 \times 16 \times 14 \times 14$ |
| $Stage_4$ | MViTv2 Block $\times$ 3 | $768 \times 16 \times 7 \times 7$ |

## 2.3 Training from Scratch vs. Training from ImageNet

As mentioned in the experiments, we do not initialize our model with ImageNet [2] pretraining due to two reasons: first, we do not assume the availability of large-scale image datasets like ImageNet or JFT [8] to ensure generalization of our method; Second, training from scratch consumes less training resources than training from ImageNet model, assuming that ImageNet-trained models are not readily available (the code and model for MViTv2 are not available at the time of writing of this paper in March 2022). Here we provide detailed analysis of the training cost:

- Training time for videos: on our GPU cluster, for Kinetics-400 (240K samples) training with 8 V100 GPUs, we are able to run at 75.2 clip/s whereas similar model variant in MViTv2 paper can achieve 91.0 clip/s (Table A.6 in MViTv2 paper). The difference may be due to different performance of I/O of clusters and video decoding methods (we use decord [2] while they use torchvision). This means that training 1 epoch of K400 takes about 3200 seconds (6900 seconds for K700) with 8xV100 GPUs.
- Training time for ImageNet: with the same 8xV100 GPUs, training on ImageNet-1K (1.28M samples) for 1 epoch takes about 1800 seconds and 11000 seconds on ImageNet-21K-P [7] (11.06M samples). All our experiments have larger than 90% GPU utilization. This roughly translates to 1/10 of the training cost **per sample** compared to video training.

Therefore, finetuning from ImageNet-1K recipe takes 34% more time (92% more if from ImageNet-21K-P) than training from scratch on K400, using our code and GPU clusters. From our experience, training from scratch will take less computation resources overall. However, finetuning from ImageNet will lead to 1-2 points top-1 accuracy boost. See Table 3. As we see, ImageNet pretraining does lead to significant improvement and we expect to see further boost with our MultiTrain method.

Table 3: Train from scratch recipe vs. train from ImageNet using MViTv2 16x4 model.

| Model | K400 Top-1/5 |
|---|---|
| MViTv2 16x4 (from scratch 200 ep) | 78.8 / 93.5 |
| MViTv2 16x4 (from IN-21K-P 75 ep) | 80.6 / 94.7 |

## 2.4 Cross-dataset Projection Weights

We provide all top-5 cross-dataset projection action pairs in Table 4.

---

[2]https://github.com/dmlc/decord

Table 4: Visualization of projection weights between different action datasets. We output the top-5 final cross-dataset projection layer weights of the K400-312p model.

| Cross-dataset Pair | Top-5 Action Pairs |
|---|---|
| K400 to MiT | [('bending metal', 'bending'), ('riding elephant', 'skating'), ('pushing wheelchair', 'swinging'), ('tossing coin', 'tattooing'), ('cleaning toilet', 'plunging')] |
| K400 to SSv2 | [('playing volleyball', 'Covering something with something'), ('washing hair', 'Moving something and something closer to each other'), ('using computer', 'Pushing something so it spins'), ('faceplanting', 'Picking something up'), ('jogging', 'Lifting something with something on it')] |
| K400 to ActNet | [('parkour', 'Capoeira'), ('running on treadmill', 'Walking the dog'), ('whistling', 'Snowboarding'), ('water sliding', 'Kayaking'), ('riding mule', 'Canoeing')] |
| SSv2 to K400 | [('Wiping something off of something', 'cleaning windows'), ('Throwing something in the air and catching it', 'dining'), ('Pretending or trying and failing to twist something', 'playing poker'), ('Turning the camera upwards while filming something', 'playing poker'), ('Pulling two ends of something so that it separates into two pieces', 'dining')] |
| SSv2 to MiT | [('Throwing something in the air and catching it', 'hunting'), ('Turning the camera downwards while filming something', 'flowing'), ('Dropping something into something', 'kneeling'), ('Pulling two ends of something so that it gets stretched', 'pulling'), ('Turning the camera upwards while filming something', 'planting')] |
| SSv2 to ActNet | [('Turning the camera downwards while filming something', 'Applying sunscreen'), ('Approaching something with your camera', 'Mooping floor'), ('Putting something upright on the table', 'Mooping floor'), ('Turning the camera upwards while filming something', 'Doing fencing'), ('Showing a photo of something to the camera', 'Playing flauta')] |
| MiT to K400 | [('gambling', 'bookbinding'), ('autographing', 'eating ice cream'), ('tearing', 'ripping paper'), ('hitchhiking', 'throwing ball'), ('sneezing', 'sneezing')] |
| MiT to SSv2 | [('drying', 'Pretending to close something without actually closing it'), ('twisting', 'Lifting up one end of something without letting it drop down'), ('fueling', 'Showing something behind something'), ('trimming', 'Putting something on the edge of something so it is not supported and falls down'), ('hitchhiking', 'Scooping something up with something')] |
| MiT to ActNet | [('fueling', 'Baton twirling'), ('pitching', 'Baton twirling'), ('snapping', 'Using parallel bars'), ('saluting', 'Baton twirling'), ('frying', 'Waxing skis')] |
| ActNet to K400 | [('Removing ice from car', 'ski jumping'), ('Playing pool', 'playing badminton'), ('Hitting a pinata', 'flying kite'), ('Tai chi', 'tai chi'), ('Putting in contact lenses', 'cutting nails')] |
| ActNet to MiT | [('Putting on shoes', 'child+singing'), ('Playing piano', 'pushing'), ('Playing racquetball', 'performing'), ('Washing face', 'drawing'), ('Baton twirling', 'feeding')] |
| ActNet to SSv2 | [('Ironing clothes', 'Pretending to be tearing something that is not tearable'), ('Tai chi', 'Throwing something'), ('Doing step aerobics', 'Pushing something so that it falls off the table'), ('Doing crunches', 'Pushing something so that it slightly moves'), ('Getting a tattoo', "Tilting something with something on it slightly so it doesn't fall down")] |