# OpenReview forum: "Multi-dataset Training of Transformers for Robust Action Recognition"
_NeurIPS.cc/2022/Conference — NeurIPS 2022 Accept_

### Official Review · Reviewer_U8Wa · 2022-06-26

**Rating:** 5
**Confidence:** 4
**Soundness:** 3 good
**Presentation:** 3 good
**Contribution:** 2 fair

**Summary:**

This paper proposes a new co-training paradigm **CrossRoad** for video representation learning.
It consists of two novel loss terms, namely informative loss and projection loss.
The informative loss encourages the variance of each dimension in the embedding to be large.
The projection loss maps predictions from other dataset heads to the current dataset class and uses ground-truth action label to compute the standard cross-entropy loss.
Experiment results show that the two auxiliary losses are helpful in co-training.

**Questions:**

1. Why removing the informative loss leads to a complete failure? Would you please provide more insights or analysis?
2. In experiments, with vanilla co-training, the K400 performance improved just a little bit (only 0.3% and is the same as MViTv2 w/o rel entry in Table 1), while performance on other datasets drops drastically. That finding is, however, quite different from the findings in CoVER. In CoVER, with vanilla joint-training, the performance on all datasets improves. What do you think could be the reason? Is it due to different pre-training or different architectures adopted? Please provide more results (like experiments with IN-21K pretraining) to support you conclusions.

**Ethics Review Area:**

["I don’t know"]

**Limitations:**

Yes.

**Strengths And Weaknesses:**

## Strength
1. This work achieves strong recognition results via co-training multiple datasets with a relatively light transformer backbone MViTv2, the improvement across multiple datasets is around 2% to 4%.
2. The efficacy of two novel components are validated via ablation study.

## Weakness
1. Large-scale image pre-training is a common practice among video transformers. However, the related results are not provided in this paper.
2. Though the authors have the ablation study for two aux losses, there exists few analyses (See questions).
3. Some experimental findings in this paper is quite different from findings in CoVER, but no explanation is provided.

---

> ### Author Response · Authors · 2022-08-02
> **Author Response to Reviewer U8Wa**
>
> Thank you for your comments and questions.
>
> **Q1: Why removing the informative loss leads to a complete failure? Would you please provide more insights or analysis?**
>
> Thanks for pointing this out. We further investigate why "-informative Loss" completely fails but "Vanilla" seems to work by running an experiment of "-informative Loss -projection add", which means we remove the projected logits addition in Eq. 5 and directly compute classification loss on the projected logits. Therefore we can consider this run as adding additional projection branches to the vanilla architecture. The results are slightly better than "Vanilla" on K400 and much better on MiT/SSv2. See Table 1* below. It indicates that adding projected logits to the original branch without informative loss would prevent the model from converging (the total loss does not go down). We will add this experiment and analysis to the paper.
>
> Table 1*:
>
> | Method | K400 | MiT | SSv2 | ActNet|
> |--------|-------|-------|-------|-------|
> | Vanilla (50ep) | 80.1 | 33.4 | 60.8 | 86.5|
> | Vanilla (200ep) | 80.6 | 35.1 | 56.8 | 86.3|
> | -Informative Loss -projection add | 80.4 | 38.7 | 62.6 | 86.5|
>
> **Q2: Vanilla joint-training compared to CoVER**
>
> Thanks for the comment. We believe the failed improvement for vanilla models are most likely due to the lack of ImageNet pretraining and a smaller model. As we have observed and stated (L270-L272), vanilla training of our model is unstable. In CoVER, a much larger backbone (121.4M parameters vs. our 51.2M parameters), higher resolution inputs (448x448 vs. our 224x224) and ImageNet-21K pretraining are used across all experiments. In Table 1*, the "-Informative Loss -projection add" has more parameters than vanilla and the performance is improved. We agree that it is important to see results with ImageNet pretraining and we are doing so in Table 2*.
>
> **Q3: Lack of ImageNet Pretraining.**
>
> As stated in response to Reviewer iVPd, training from scratch is about twice as fast as the training recipe with ImageNet-21K pretraining. We did not conduct ImageNet pretraining due to limited resources (and MViTv2 authors had not released code or pretrained models at the time of experiments). We have downloaded ImageNet-21K-P [7*] and conducted pretraining experiments on K400 with a smaller 16x4 model (we used 32x3 model in the paper) as shown in Table 2*. We are still running the CrossRoad method with K400-multi-dataset but may take 10 days to finish (due to significant budget cut in GPU spending, we are running with 4xV100 GPUs whereas before we had 128). As we see, ImageNet pretraining does lead to significant improvement and we are expecting to see further boost with our CrossRoad method. We will add the experiments with 32x3 models in the revised version.
>
> Table 2*: ImageNet pretraining
>
> | Model | K400 |
> |--------|-------|
> | MViTv2 16x4 (from scratch 200 ep) | 78.8 / 93.5 |
> | MViTv2 16x4 (from IN-21K-P 75 ep) | 80.6 / 94.7 |

---

### Official Review · Reviewer_iVPd · 2022-07-06

**Rating:** 5
**Confidence:** 4
**Soundness:** 3 good
**Presentation:** 3 good
**Contribution:** 2 fair

**Summary:**

The paper proposes a method to train video transformer on multiple video datasets. Instead of simply applying multiple cross-entropy loss over, the authors proposed to manipulate on embeddings encoded by an improved multiscale vision transformer to capture the intrinsic relations between classes across different action datasets. Specifically, they first adopt the informative loss from Barlow-Twins to maximize variance and covariance across embedding channels. Second, they propose to perform directed project from one dataset's classification head to another, to learn the label relation across datasets. The results show that the full pipeline is superior to vanilla training over multiple datasets and achieve state-of-the-art result.

**Questions:**

1. "Note that our model does not use any image training datasets, and our model computation cost is only a fraction of the baselines" (L247-248). It is true that pre-training on large image datasets. However, training video model from scratch typically means more epochs for convergence. (e.g. MViTv2 uses 200 epochs). Since training image model is often cheaper (1/10 than video model), I would see more discussion to justify this argument.

2. MViTv2 without Rel-PE. If I am not mistaken, MViTv2 uses relative positional embedding by default (See their Figure 2). Therefore I do not fully understand the statement "The “MViTv2 w/o rel” indicates the model without the relative positional embedding in the original paper" (L234-235).


====
Post-rebuttal revision:
My questions have been addressed. A 34-90% improvement over image-based pretraining at the cost of 1-2% drop of accuracy is fine but not impressive. The learned directed class projection is somewhat interesting. Therefore I will keep the rate of "5 Borderline accept".

**Limitations:**

Yes. The authors have discussed the limitations, e.g., co-training is limited on video datasets only. They have also covered potential negative societal impacts such as dataset biases.

**Strengths And Weaknesses:**

### Strengths

+ The performance is good considering that CrossRoad is purely trained on top of video dataset without any image pretraining.

+ The method is simple and the informative loss borrowed from BarlowTwins seems effective, especially when we want to include projection across different classification heads.

### Weaknesses

- Cross-dataset Training is not only a standalone problem in the video domain. There has been a few works on other tasks such as detection [1,2], segmentation [3]. The authors are suggested to consider discussing these methods in the related work for broader scope.

[1] Zhou, Xingyi, et al. "Simple multi-dataset detection." CVPR 2022.
[2] Wang, Xudong, et al. "Towards universal object detection by domain attention."CVPR 2019.
[3] Lambert, John, et al. "MSeg: A composite dataset for multi-domain semantic segmentation." CVPR 2020.

- A few arguments need further justification, including
  - The computation cost comparison between training video model from scratch v.s. pre-training on image dataset and then finetuning. (See Q1.)
- Some settings in the ablation need further clarification, including:
  - The reason of studying MViTv2 without relative position embedding (Q2.).
  - "Vanilla" cross-dataset training v.s. "- informative loss". I don't fully get the difference. Do you mean "Vanilla" = CrossEntropy (CE) only, "- informative loss" = CE + projection loss?

- What is $\sigma_k$ in Eq (7) and L198 and how do you determine the value?

- How does the learned directed class projection weights look like? Some visualization and discussion might be preferred.

- How do you construct the video clips on ActivityNet? Do you uniformly cut the entire video into 10-second clips or only keep those temporal segments annotated as activities. This might be useful for later efforts trying to reproduce the results.

---

> ### Author Response · Authors · 2022-08-02
> **Author Response to Reviewer iVPd**
>
> Thank you for the helpful comments and suggestions. We will add suggested cross-dataset/multi-dataset training methods to our related work. Here we answer the specific questions below.
>
> **Q1: Clarification on computation cost (from scratch on videos vs. using large-scale image datasets).**
>
> Thank you for your comment. The computation cost on L248 we refer to is the inference cost (the "gFLOPs" column in Table 1 & 2), not training cost. We will clarify this point in the paper.
>
> As for training cost, the reviewer is right that for MViTv2 or other similar ViT models, their image versions with the same spatial resolution are usually 1/10 or even 1/20 of their video versions in terms of inference FLOPs. Below we compare training wall time (since this directly converts to money spending on GPU clusters) of different setups.
>
> For MViTv2, the training schedule for image-initialized models are 300 epochs on ImageNet-1K (or 90 epochs on ImageNet-21K+ImageNet-1K) and then fine-tune on Kinetics for 100 epochs (or 75 epochs for ImageNet-21K models). Please refer to Appendix B.2 and B.4 in the MViTv2 paper for more details. Previously we did not use ImageNet pretraining mainly because we did not have a copy of the ImageNet data (1.3TB for ImageNet-21K!). We now have downloaded ImageNet-1K and ImageNet-21K-P [7*] (a compressed version of ImageNet-21K) for training cost experiments.
> 1. Training time for videos: on our cluster, for Kinetics-400 (240K samples) training with 8 V100 GPUs, we are able to run at 75.2 clip/s whereas similar model variant in MViTv2 paper can achieve 91.0 clip/s (Table A.6 in MViTv2 paper). The difference may be due to different performance of I/O of clusters and video decoding methods (we use decord [8*] while they use torchvision). This means that training 1 epoch of K400 takes about 3200 seconds (6900 seconds for K700) with 8xV100 GPUs.
>
> 2. Training time for ImageNet: with the same 8xV100 GPUs, training on ImageNet-1K (1.28M samples) for 1 epoch takes about 1800 seconds and 11000 seconds on ImageNet-21K-P (11.06M samples). All our experiments have >90% GPU utilization. This roughly translates to 1/10 of the training cost **per sample** compared to video training.
> Therefore, finetuning from ImageNet-1K recipe takes 34% more time (92% more if from ImageNet-21K-P) than training from scratch on K400, using our code and GPU clusters. From our experience, training from scratch will take less computation resources overall. However, finetuning from ImageNet will lead to 1-2 points top-1 accuracy boost (See Table 2* in response to Reviewer U8Wa).
>
> **Q2: MViTv2 without relative positional embedding.**
>
> Thanks for pointing this out. We implemented MViTv2 based on their paper and the MViTv1 code [9*]. We were not able to implement the relative positional embedding part due to missing details (symmetric relative or not, etc.) in their paper so we use decomposed absolute positional embedding. The official code for videos has not released until July [9*]. Therefore we experiment with the "MViTv2 w/o rel" setting for all our runs (L137 and footnote). The number of "MViTv2 w/o rel" comes from Table A.6 of the MViTv2 paper (it should be 80.4 instead of 80.1, for MViTv2 w/ abs. pos. We will fix it in the revised version) as a reference. However, our baseline can only achieve 79.8 (Table 1) with the same training recipe, which may be due to slight differences in the Kinetics dataset (missing some videos, etc.). We will make it clear in the edition.
>
> **Q3: "Vanilla" cross-dataset training v.s. "- informative loss".**
>
> Yes, you are right that "Vanilla" = CrossEntropy (CE) only, "- informative loss" = CE + projection loss. The full model uses both losses. We will revise Table 3 to make it clear.
>
> **Q4: What is \sigma_k in Eq (7) and L198 and how do you determine the value?**
>
> \sigma_k is a scalar that works as a weighting term for dataset k. \sigma is a vector of learnable parameters to avoid tuning loss weights for different datasets (L196). We will make it clear.
>
> **Q5: ActivityNet dataset construction.**
>
> We cut the annotated segments of the videos into 10-second long clips and split the dataset into 107K training and 16K testing (L214 - L215). We will make it clear and release the exact data preparation code.
>
> **Q6: Visualization of projection weights.**
>
> Thanks for the suggestion. We are in the process of generating the visualization and will update it in the revised version.
>
> [7*] Ridnik, Tal, et al. "Imagenet-21k pretraining for the masses." NeurIPS 2021.
>
> [8*] https://github.com/dmlc/decord
>
> [9*] https://github.com/facebookresearch/SlowFast

---

> > ### Author Response · Authors · 2022-08-05
> > **Update to Q6**
> >
> > As suggested, we analyze the cross-dataset projection weights of the K400/312p model in Table 1, listed top 5 concepts for each pair of datasets below.
> > We make two observations:
> > 1. The top projections are visually similar actions, which confirms our intuition that there are intrinsic relations in the datasets that the model can mine to improve performance. For example, “bending metal” in K400 and “bending” in MIT, “parkour” in K400 and “Capoeira” in Activitynet. Interestingly, “Wiping something off of something” in SSv2 and “cleaning windows” in K400.
> > 2. The action with the same name may not have the highest weights. In “mit to kinetics”, the “sneezing” action ranks 5th in the projection weights, suggesting that there might be discrepancies of the same concept in different datasets.
> >
> > These observations are interesting and one may compare the learned weights with textual semantic relations (like those in ConceptNet). We leave this to future work. We will add this analysis to the revised version.
> >
> > kinetics to mit
> > [('bending metal', 'bending', 0.19999029), ('riding elephant', 'skating', 0.19910285), ('pushing wheelchair', 'swinging', 0.19789538), ('tossing coin', 'tattooing', 0.1961417), ('cleaning toilet', 'plunging', 0.19546732)]
> > ----------------------------------------
> > kinetics to ssv2
> > [('playing volleyball', 'Covering something with something', 0.1490408), ('washing hair', 'Moving something and something closer to each other', 0.14768724), ('using computer', 'Pushing something so it spins', 0.1392663), ('faceplanting', 'Picking something up', 0.13809656), ('jogging', 'Lifting something with something on it', 0.13718487)]
> > ----------------------------------------
> > kinetics to activitynet
> > [('parkour', 'Capoeira', 0.1922843), ('running on treadmill', 'Walking the dog', 0.17684066), ('whistling', 'Snowboarding', 0.17329654), ('water sliding', 'Kayaking', 0.16401285), ('riding mule', 'Canoeing', 0.16242917)]
> > ----------------------------------------
> > mit to kinetics
> > [('gambling', 'bookbinding', 0.1396648), ('autographing', 'eating ice cream', 0.13093388), ('tearing', 'ripping paper', 0.12916225), ('hitchhiking', 'throwing ball', 0.12469659), ('sneezing', 'sneezing', 0.12128164)]
> > ----------------------------------------
> > mit to ssv2
> > [('drying', 'Pretending to close something without actually closing it', 0.11436209), ('twisting', 'Lifting up one end of something without letting it drop down', 0.11326913), ('fueling', 'Showing something behind something', 0.11158158), ('trimming', 'Putting something on the edge of something so it is not supported and falls down', 0.10773694), ('hitchhiking', 'Scooping something up with something', 0.10572156)]
> > ----------------------------------------
> > mit to activitynet
> > [('fueling', 'Baton twirling', 0.12002337), ('pitching', 'Baton twirling', 0.11702621), ('snapping', 'Using parallel bars', 0.11015125), ('saluting', 'Baton twirling', 0.10861136), ('frying', 'Waxing skis', 0.10808712)]
> > ----------------------------------------
> > ssv2 to kinetics
> > [('Wiping something off of something', 'cleaning windows', 0.13989474), ('Throwing something in the air and catching it', 'dining', 0.13834158), ('Pretending or trying and failing to twist something', 'playing poker', 0.13351966), ('Turning the camera upwards while filming something', 'playing poker', 0.13347316), ('Pulling two ends of something so that it separates into two pieces', 'dining', 0.13303888)]
> > ----------------------------------------
> > ssv2 to mit
> > [('Throwing something in the air and catching it', 'hunting', 0.1650513), ('Turning the camera downwards while filming something', 'flowing', 0.16386871), ('Dropping something into something', 'kneeling', 0.16295567), ('Pulling two ends of something so that it gets stretched', 'pulling', 0.16081172), ('Turning the camera upwards while filming something', 'planting', 0.15848677)]
> > ----------------------------------------
> > ssv2 to activitynet
> > [('Turning the camera downwards while filming something', 'Applying sunscreen', 0.12933515), ('Approaching something with your camera', 'Mooping floor', 0.12181736), ('Putting something upright on the table', 'Mooping floor', 0.11953914), ('Turning the camera upwards while filming something', 'Doing fencing', 0.118271396), ('Showing a photo of something to the camera', 'Playing flauta', 0.11793875)]
> >
> > (omitting activitynet’s due to space limit)

---

> > > ### Author Response · Authors · 2022-08-08
> > > **Kind Reminder**
> > >
> > > Dear Reviewer iVPd,
> > > Thank you very much again for the time and effort put into reviewing our paper. We believe that we have addressed all your concerns in our response. We have also followed your suggestion to improve our paper and have added additional experimental analysis. We kindly remind you that we are approaching the end of the discussion period. We would love to know if there is any further concern, additional experiments, suggestions, or feedback, as we hope to have a chance to reply before the discussion phase ends.

---

> > > > ### Comment · Reviewer_iVPd · 2022-08-09
> > > > **Reply to the authors**
> > > >
> > > > Dear authors,
> > > >
> > > >  Thank you for the clarification. My concerns have been addressed and I don't have further questions.

---

### Official Review · Reviewer_YQNQ · 2022-07-11

**Rating:** 5
**Confidence:** 4
**Soundness:** 2 fair
**Presentation:** 2 fair
**Contribution:** 2 fair

**Summary:**

This paper is the first work to propose an informative representation of regularization into cross-dataset action recognition. This work makes full use of existing visual transformer backbones. This method is dedicated to learning robust and informative representations. By combining the projection loss, this work can effectively mine intrinsic class relations. Experiments on different datasets prove the method can achieve better performance and produce state-of-the-art results.

**Questions:**

- The table caption should be above the column.
- The cross-dataset action recognition has yielded many forms such as Video Domain Adaptation and Video Domain Generalization. The cross-dataset in this paper focuses more on multi-domain learning. The concept of cross-dataset in this paper may lead to misleading and need to be clarified.
- This paper should add the multi-domain baselines for a fair comparison with the existing multi-domain methods.
- The author should clarify the novelty of this paper as described in Strengths And Weaknesses.


======
I have read the author's comments. Most of my concerns are clarified. I will increase my score.

**Limitations:**

Yes

**Strengths And Weaknesses:**

Strengths:
- This work seems to be the first work to introduce informative representation regularization into cross-dataset training for action recognition. It explores how to learn robust representations among multiple video domains.
- To my knowledge, this work is the first work to bring the informative loss and projection loss into cross-dataset action recognition. This self-supervised loss can bring performance gains.
-  This method may be suitable for any action recognition model.

Weakness:
- This work seems to be the combination of existing video backbones, projection loss, and informative loss. Cross-dataset action recognition is a challenging problem due to the temporal information inner the video sequence. This paper did not consider more on the temporal information in the sequence. The design of the projection loss and the informative loss did not consider the temporal dynamics. The projection loss and the informative loss should be carefully designed for this specific cross-dataset action recognition task not directly used.
- This work has no contributions to both the backbone and loss. The novelty of this work should be clarified.
- The problem this work studies is a multi-domain problem. However, this work did not compare with multi-domain methods.

---

> ### Author Response · Authors · 2022-08-02
> **Author Responses to Reviewer YQNQ**
>
> Thanks for the helpful comments and suggestions. We will add all the mentioned related work and address the table caption format problem in the revised version.
>
> In this paper, our practical goal is to propose a training paradigm for parameter-efficient models across multiple datasets, which could lead to less inference time overall for recognizing the same number of action classes.
> Here we address specific questions below.
>
> **Q1: Comparison with video domain generalization and multi-domain methods.**
>
> A1: Sorry for the confusion. Our work is under the multi-task learning topic, not video domain generalization (DG). The key distinction is that our goal is to train a single model on multiple related tasks (multiple action datasets) such that the model performs well on the same set of tasks, whereas DG aims to generalize a model to unseen data distributions. Please refer to Table 2 of this latest published survey on domain generalization [1*].
>
> In terms of datasets and experimental setting of video domain generalization, we have found that most DG methods [2*, 3*, 4*, 5*] are compared on the UCF-HMDB dataset, and they mostly followed the adversarial domain generalization framework [6*]. Specifically, our setting differs in that:
>
> 1. the size of the datasets. In video DG, the training and testing datasets usually contain a couple of thousand videos, whereas our setting includes large-scale video datasets of millions of videos.
>
> 2. number of action classes. In video DG, it usually requires that the target action classes are shared in both source and target domains. Hence the action classes involved are usually less than 14 (See Table 1 in [5*]). In our setting, which is the same or similar as in [27, 45, 1, 11], we aim to train a single-model for over a thousand action classes (and the classes need not to be shared across datasets).
>
> 3. training objectives. In both single-source or multi-source video DG, methods aim to learn a model in such a way that the model can generalize well to any Out-of-Distribution target domain. We train and test on the same domains. In a way our setting is easier than video DG (as well as video domain adaptation, where target domain data is sparsely provided) since we have access to the target domain data and labels during training. In terms of model performance, in a recent work [5*], under DG setting, top-1 accuracy on Kinetics (14 classes) is under 20% while our method averages 80%+.
>
> We apologize for using the misleading "cross-dataset" term. We will revise our phrasing (changing to "co-training", etc.), make the task distinction clear in Section 1 & 2 and add the aforementioned references. Please also provide specific references if the reviewer thinks we are still missing any.
>
> **Q2: Clarify novelty of this paper on backbone, loss and temporal modeling.**
>
> A2: We respectfully disagree with the statement that "this work has no contributions to both the backbone and loss". We clarify our contributions below.
>
> Our method is closely related to the line of multi-task learning methods [1, 27, 11, 45] for videos. Previous works of this field are scarce maybe due to the large demand for computation resources. In [1] (NeurIPS'21) the authors proposed a multi-modal transformer and a novel data augmentation method for training. In [45] and [27] the authors proposed to train with both image and video or other tasks simultaneously to improve performance. In [11] (ECCV'20) the authors proposed to train a teacher model to filter webly-labeled data for the final omni-source training.
>
> Our method is the first to propose a simple yet effective way (no multi-stage training, no complex dataset-wise sampling, no dataset-wise hyper-parameter tuning) to capture multi-action relations (L183-L187) and informative representations using SOTA vision transformers (L30-L57).
>
> We are the first to effectively combine the informative loss (inspired by self-supervised contrastive learning in image recognition [3]) and the multi-task projection loss (built upon multi-task learning in image domain [22]) to provide a principled way for multi-action-dataset training (Section 3.2).
>
> The proposed method has been deployed in industrial products and it is proven to save computation resources in practice.
>
> In this work, we do not put emphasis on temporal modeling due to the fact that the base model (MViTv2) only has a receptive field of 2-3 seconds temporally. We leave long-term multi-action recognition for future work. We will clarify this part.
>
> [1*] "Domain generalization: A survey." TPAMI 2022.
>
> [2*] "VideoDG: generalizing temporal relations in videos to novel domains." TPAMI 2021.
>
> [3*] "Temporal attentive alignment for large-scale video domain adaptation." ICCV 2019.
>
> [4*] "Shuffle and attend: Video domain adaptation." ECCV 2020.
>
> [5*] "Dual-Head Contrastive Domain Adaptation for Video Action Recognition." WACV 2022.
>
> [6*] "Adversarial discriminative domain adaptation." CVPR 2017.

---

### Meta-Review · Area_Chair_dsZQ · 2022-08-26

**Recommendation:** Accept
**Confidence:** Certain

**Metareview:**

The paper proposes a co-training method for video representation learning, by training video transformers on multiple video datasets. The paper proposes two novel loss terms: informative loss and projection loss. The informative loss encourages the variance of each dimension in the embedding to be large. The projection loss maps predictions from other datasets to the current dataset,  to learn the label relation across datasets by using ground-truth action labels to compute standard cross-entropy loss.

Based on the feedback provided by the reviewers, we recommend this paper for publication at NeurIPS 2022.

The reviewers had some concerns about the paper. Reviewer YQNQ had concerns that the design of the projection loss and the informative loss did not consider the temporal dynamics, and that it does not compare with multi-domain methods. Reviewer iVPd  recommended considering tasks like detection, segmentation, etc.  discussing these methods in the related work for broader scope. Reviewer U8Wa mentioned that the experimental findings in this paper are quite different from findings in CoVER, but no explanation is provided.

We thank the authors for addressing the comments of the reviewers in their review during the author feedback period. The authors seem to have addressed some of the concerns/feedback from the reviewers with detailed discussions -- it would be good to include these discussions, as much as possible, in the updated paper or supplemental materials.

**Award:**

No

---

### Decision · Program_Chairs · 2022-09-14

Accept